# The role of iron deficiency and factors associated with anemia during pregnancy in Southeastern Tigray, Ethiopia, 2020

Tedros Bereket[ID]*, Freweini Gebrearegay Tela, Gebretsadkan Gebremedhin Gebretsadik, Selemawit Asfaw Beyene

Department of Nutrition and Dietetics, School of Public Health, College of Health Sciences, Mekelle University, Tigray, Ethiopia

* Tedros.bereket99@gmail.com

**Data Availability Statement:** All relevant data are within the manuscript and its Supporting Information files.

## Abstract

### Background

Pregnant women are more likely to experience anemia due to their increased need for nutrients, and anemia has been associated with unfavorable maternal-fetal outcomes. In Ethiopia, anemia rates are increasing despite efforts to reduce them. Besides, the extent to which iron deficiency contributes to anemia in this population is unclear. Therefore, this study aimed to assess the role of iron deficiency and factors associated with anemia during pregnancy in the southeastern zone of Tigray, Ethiopia.

### Method

A facility-based, cross-sectional study was conducted among 311 pregnant women who attended four health facilities in the southeastern zone of Tigray region from January to June 2020. The study utilized a semi-structured, pretested questionnaire to collect data. The data were entered into Epi data 3.1 and analyzed using SPSS version 25. Candidate variables with a p-value $\leq 0.25$ in the bivariate logistic regression were considered to the final model of multivariate logistic regression analysis. Adjusted odds ratio with a 95% confidence interval and a p-value of $\leq 0.05$ were used to declare statistical significance for the factors associated with anemia.

### Results

A total of 311 mothers had completed the survey with a response rate of 98.1%. The overall magnitude of anemia was 55(17.7%). Out of the anemic pregnant women, 30(54.5%) were due to iron deficiency. Third-trimester pregnancy (AOR = 4.9; 95% CI:1.49, 15.98), inter-pregnancy gap of $\leq 2$ years (AOR = 7.7; 95% CI: 2.71, 21.64), coffee/tea intake immediately after meal (AOR = 3.0; 95% CI: 1.02, 8.92), low diet diversity score (AOR = 4.2; 95% CI: 1.02,17.57), not taking iron-folic acid supplementation (AOR = 2.6; 95% CI: 1.05, 6.70), and Mid upper arm circumference value of < 23cm (AOR = 2.7; 95% CI: 1.16, 6.47) were independent factors associated with anemia.

**Funding:** The study was supported by NORAD project. The funder has no role in the study design; field management; collection, analysis, and interpretation of data; training of staff; report writing, and manuscript publication. The funders had no role in study design, data collection and analysis, decision to publish, or preparation of the manuscript.

**Competing interests:** The authors have declared that no competing interests exist.

## Conclusion

Based on the findings, anemia is a mild public health concern in the study areas. Additionally, over 50% of the anemia cases were attributed to iron deficiency anemia. Nutritional counseling on food diversification, especially the consumption of iron-rich foods, and delaying the drinking of coffee/tea after a meal, as well as iron-folic acid supplementation, are needed to reduce anemia in the study area.

## Introduction

Anemia is a condition where the level of hemoglobin (Hg) in the body is lower than the normal range due to a prolonged negative iron balance [1]. World Health Organization defines anemia during pregnancy as having a Hg concentration of less than 11.0 g/dl in the first and third trimesters, and less than 10.5 g/dl in the second trimester [2]. The etiology of anemia is multifactorial and the most frequent causes of anemia include nutritional deficiencies of iron, other micronutrient deficiencies, chronic communicable and non-communicable diseases, and genetically inherited blood disorders [3]. Among these causes, iron deficiency is often the primary contributor to anemia. However, the extent of this contribution can vary by population, region, and environmental factors [4]. Iron deficiency anemia (IDA) is characterized by low hemoglobin and ferritin concentration. It occurs when the intake of total or bioavailable iron is inadequate to meet the body's iron demands or to compensate for increased losses, and it represents the final stage of iron depletion [4, 5].

Anemia in pregnancy is a global public health problem affecting both developing and developed countries with major consequences for human health as well as social and economic development [1, 6]. According to the WHO report, it is estimated that 38% of pregnant women have anemia globally corresponding to 32 million pregnant women. The prevalence was high in developing countries like Africa accounting for 46.3% and Southeast Asia 48.7%, the global prevalence estimate that the prevalence among pregnant mothers in sub-Saharan countries was high [7].

Pregnant women are more prone to anemia particularly to iron deficiency anemia because iron requirements increase during pregnancy and are difficult to cover by diet alone [4]. This vulnerability is attributed to the elevated iron needs during pregnancy, which predisposes pregnant women to the development of iron-deficiency anemia [8]. During pregnancy, Anemia is linked to various adverse outcomes for both mother and infant, including an increased risk of hemorrhage, sepsis, maternal mortality, perinatal mortality, and low birth weight [9]. According to WHO data, anemia is associated with 40% of maternal deaths worldwide, and IDA is an underlying risk factor for 115,000 of the 510,000 maternal deaths (i.e. 22%) and 591,000 of the 2,464,000 perinatal deaths (i.e. 24%) occurring annually around the world [7].

Despite there have been numerous efforts to address anemia, it continues to pose a persistent challenge for pregnant women. Shockingly, it is not decreasing at the same rate as other nutritional issues [7]. As a developing country, Ethiopia faces the challenge of high rates of anemia. Recognizing the impact of this condition, the government has implemented various initiatives such as administering deworming treatments, distributing insecticide-treated bed nets (ITN) to prevent malaria, and increasing the coverage of antenatal care (ANC) follow-up [7].

However the Ethiopian Demographic and Health Survey (EDHS) 2016 report revealed a concerning trend: the prevalence of anemia among pregnant women has risen from 22.4% in 2011 to 29.1% in 2016. In the Tigray region, this increase is even more pronounced, with rates

rising from 12.4% in 2011 to 19.7% in 2016 [8, 9]. Access to local data on anemia and iron deficiency is essential to effectively manage and control anemia in pregnancy. Unfortunately, despite the upward trend in anemia rates, information on the contribution of iron-deficiency anemia to total anemia in the Tigray region is currently limited. To address this gap, our study aimed to determine the role of iron deficiency anemia to total anemia, and identify factors associated with anemia among pregnant mothers in southeastern Tigray, Northern Ethiopia

## Materials and methods

### Study area, period, and population

The study was conducted among 311 pregnant women who came for ANC from January to June 2020 in public health facilities of southeastern zone of the Tigray region, northern Ethiopia. The southeastern zone is one of seven zones located within the Tigray region, with Southern Zone to the south, the Amhara Region to the southeast, Central Zone to the northeast, Eastern Zone to the north, and Afar Region to the east. Additionally, the Mekelle Special Zone is surrounded by the Southeastern zone. Two districts, Degua Tembien and Hintalo Wajrat, were chosen for the investigation using the simple random sampling method. Degua Tembien, located 2625 meters above sea level and situated 50 km from the regional capital of Mekelle, is comprised of 23 tabias. Hintalo Wajrat, on the other hand, is found in the southeastern part of Tigray and has an elevation of 2100 meters above sea level, with 22 Kebeles within the district.

### Study design

A facility-based cross-sectional study design was employed.

### Inclusion and exclusion criteria

Pregnant women who had been residents of southeastern zone at least six months before the study, attended the selected public health facilities in southeastern zone for ANC follow up during the study period and were willing to participate were included, and pregnant women who were critically ill, unable to hear and revisited the ANC for follow up during the study period were excluded from the study.

### Sample size calculation

The sample was calculated using epi-info version—7 through a single population proportion with the assumption of 95% confidence interval, 5% margin of error, and considering 29.1% proportion anemia among pregnant women from the EDHS 2016 report [9]. Initially, the calculated sample size was 318. However, due to the population of pregnant women in the south eastern zone with ANC follow up being fewer than 10000 mothers according to the 2018/2019 report, a correction formula was applied, resulting in a final sample size of 288. Finally, by considering the 10% non-response rate the final sample size was 317.

### Sampling technique

The southeastern zone encompasses four districts: Degua Tembien, Enderta, Hintalo Wajrat, and Saharti Samre. Among these, two districts were chosen at random: Degua Tembien and Wajrat. Degua Tembien has six health facilities, and Hintalo Wajrat has seven health facilities. Four health facilities consisting of two primary hospitals and two health center named Hagere selam (n = 103), Adigudom(n = 101), Alasa(n = 54), and Hiwane(n = 59) were randomly selected from the two chosen districts, with two from each. Study participants were selected through proportional allocation based on the number of pregnant women at each respective

health center or primary hospital. Pregnant women were consecutively chosen until the total sample size was attained.

## Study variables

**Dependent variable**: Anemia.

**Independent variables. Socio-economic and demographic**: maternal age, marital status, religion, occupational status, educational status, family size, and income.

**Nutrition-related**: food security, DDS, IFA supplementation, meal frequency, food taboo, coffee/tea consumption after meal and MUAC.

**Pregnancy-related**: parity, length of gestation, number of ANC contacts, interpregnancy gap, history of abortion, menstrual bleeding, and birth spacing.

**Disease-related**: hookworm, malaria, and HIV.

## Data collection

During the data collection process, an interviewer-administered semi-structured questionnaire was used. In regards to anthropometric measurements, the MUAC measurement was taken twice on each pregnant woman's non-dominant hand using non-stretchable MUAC tape, with no cloth, at the mid-point between the tips of the shoulder and elbow. This was done to ensure accuracy and recorded to the nearest 0.1 cm. For DDS data, a 24-hour recall individual dietary diversity questionnaire was utilized. The pregnant woman was asked to recall all food items consumed in the previous 24 hours leading up to the survey date. If there was a special occasion the day before, the respondent was asked about foods consumed before that. The reported food items were then classified based on ten food groups. As a cut-off point, a mother was defined as having "poor dietary diversity" if she consumed less than 5 food groups and "good dietary diversity" if she consumed 5 or more food groups [10]. Measuring food security status was adopted from Food and Nutrition Technical Assistance (FANTA) Household Food Insecurity Access Scale guidelines. The questionnaire usually contains 18 questions; the first nine questions were answered with yes or no after respondents were asked to recall whether the condition in each occurrence question happened at all in the past four weeks or not. If the answers were "yes" to an occurrence question, a frequency of occurrence question was asked to determine whether the condition happened rarely (once or twice), sometimes (three to ten times), or often (more than ten times) in the past four weeks. These scores were classified into one of the four categories: food secured, mildly food insecure, moderately food insecure, and severely food insecure [11].

To evaluate the level of anemia knowledge in pregnant women, a set of ten questions related to the condition were posed to all participants. Based on their responses, pregnant women were categorized as having good knowledge if their score was equal to or greater than the mean value of correct answers, while those with scores below the mean were deemed to have poor knowledge. The questionnaire used to gather socio-demographic and other anemia-related factors in pregnant mothers was developed through a review of various literature sources [11–16].

**Laboratory methods.** Hemoglobin concentration was measured using a hemoglobin-ometer (HemoCue HB 201+ analyzer, SETEMA Limited PLC, Ängelholm, Sweden). The participant's fingertip was cleansed with an alcohol swab, and a lancet was used to puncture the selected finger. The first and second drops of blood were discarded, and the third drop (10μl) was collected in a micro cuvette. Any excess blood was wiped off, and the micro cuvette was then inserted into the analyzer for hemoglobin concentration measurement. Finally, the puncture site was cleaned with an alcohol swab. The altitude was taken into account before defining

anemia, with 0.8 subtracted for Hintalo wajrat district (2100m above sea level) and 1.3 subtracted for Degua tembien district (2625m above sea level). Pregnant women with hemoglobin level 11 g/dl in the first and third trimesters, and hemoglobin (Hb) < 10.5 g/dl in the second trimester were classified as anemic. Severe, moderate, and mild anemia was defined as Hb below 7gm/dl, 7–9.9 g/dl, and 10–10.9 g/dl respectively.

Pregnant women who had a hemoglobin level of less than 10.5 g/dl during their second trimester, and less than 11 g/dl during their first and third trimesters, were identified as anemic and were given a serum ferritin test. A sterile, disposable plastic syringe was used to draw approximately 3 ml of venous blood from the antecubital vein, which was then collected into serum-separating tubes. After centrifugation in the health center laboratory, the serum was separated and kept at -20˚C in a refrigerator. The sample was then transported under appropriate conditions, using a dry ice box, to an international clinical laboratory located in Mekelle for further analysis. Ferritin reagent, ferritin immune control high/low, and ferritin calibrator reagents were used to measure ferritin levels. It is important to note that serum ferritin is an acute-phase protein that can show false normal values or be elevated during inflammation or infection. To rule out the presence of inflammation or infection, C-reactive protein measurement was used as an early marker. Pregnant women who had a CRP level of ≤5mg/l and serum ferritin levels of less than 15 µg/L, or a CRP level of >5mg/l and serum ferritin levels ≤30, were referred to have iron-deficiency anemia.

**Data quality assurance.** To maintain the quality of data, four public health officer (B. Sc.) data collectors and one first-year nutrition graduate supervisor were recruited and trained for two days to have a common understanding of how to take informed consent, how to approach participants, ethical procedure, and how to take an anthropometric measurement. The questionnaire was initially prepared in English, translated to Tigrigna, and then back-translated to ensure consistency of concepts. Before the actual data collection, the questionnaire was pre-tested for validation with 5% of the study population in Tikul health center which was not included in the study, and Findings and experiences from the pretest were utilized in modifying the data collection tool before the data collection process. Blood sample collections and laboratory procedures were taken by senior experienced laboratory technicians who have a background in health. To achieve the highest level of accuracy and reliability all processes and procedures in the laboratory were performed in the best possible way, and safety precautions were followed at all points in the testing process from specimen collection to testing, storage, and disposal of biohazard wastes, blood samples were transferred by dry ice-box under appropriate conditions to avoid exposure to high temperature.

## Data analysis

The questionnaire was coded, and data were entered using EPI data 3.1 and later exported to SPSS version 25 for cleaning and analysis. Normality for the distribution of continuous variables was checked with a histogram, for normally distributed data, the mean and standard deviation (SD) were reported, and for non-normally distributed data, the median and interquartile range were reported.

Bivariate logistic regression analysis was carried out to select explanatory variables that fit the final model at a p-value of 0.25 and a confidence interval (CI) of 95%. Finally, all variables with a P value of less than 0.25 in the bivariate analysis were entered into multivariate analysis to control the possible effect of confounders. Multivariate logistic regression analysis was employed to identify factors associated with anemia, and an adjusted odds ratio (AOR) with a corresponding 95% confidence interval was computed to show the strength of

association, and a *P*-value of <0.05 was used to declare statistical significance. Multicollinearity was checked using variance inflation factor (VIF) and it lies between 1 and 10 for all of the independent variables. Hosmer and Lemshow test (0.45) for the fitness of the model was also done.

## Operational definitions

- Iron deficiency anemia: serum ferritin level <15 μg/L in the absence of inflammation, serum ferritin <30 μg/Lin presence of inflammation [17].

- Anemia during pregnancy: pregnant women with hemoglobin level < 11 g/dl in the first, and third trimesters and hemoglobin < 10.5 g/dl in the second trimester, adjusted for altitude [18].

- Mild anemia: Hemoglobin concentration level between 10–10.9 g/dl for pregnant women.

- Moderate anemia: Hemoglobin concentration level between 7.0–9.9 g/dl for pregnant women

- Severe anemia: Hemoglobin concentration level <7.0 g/dl for pregnant women. Low women dietary diversity score: when a pregnant mother consumed <5 food groups [10].

- High women dietary diversity score: when a pregnant mother consumed ≥5 food groups [10].

- Under nutrition: Nutritional status of pregnant women with MUAC measurement < 23cm. Adherence to IFA supplementation: low adherence (score < 6), medium (scores of 6 & 7), and high (score of 8) [19].

- Good knowledge about anemia:—pregnant women who scored greater than the median value of the correct response.

- Poor knowledge about anemia:—pregnant women who scored lower than the medianvalue of the correct response.

## Ethical considerations

The study was conducted with the proper ethical approval from the Mekelle University College of Health Science Institutional Review Board (IRB) (reference number MU/ERC 1541/2020, approval dated 20/02/2020). Additionally, a support letter was obtained from TRHB, southeastern zone health office, and respective health institutions in the study area. Before conducting the actual data collection, each participant was thoroughly informed of the objective and purpose of the study and provided with oral consent in a private room after receiving their routine ANC services. Throughout the study, confidentiality and privacy were strictly maintained, and participants were guaranteed the right to participate, refuse, or stop at any time during the data collection process.

To minimize risk during blood sample collection, laboratory technicians were informed to follow the standard operating procedure (SOP) and blood was drained by senior laboratory professionals. To ensure confidentiality, names, and other means of identity were not used during the data collection. Any result of the test that was relevant to the participant was communicated immediately and linked to the health facility for treatment and follow-up.

## Result

### Socio-demographic characteristics of study participants

A total sample consisting of 311 pregnant women with a response rate of 98.1% participated in this study. The age of the pregnant women ranged from 17 to 38 with a mean age ± SD of 26.8 ± 5.3. Two-hundred ninety-three (94.2%) pregnant women were orthodox Christian followers and 288 (92.6%) of them were married. Regarding the educational status of the participants, 109 (35%) of them were unable to read and write and only 18 (5.8%) of them had attained college and above (Table 1).

### Pregnancy-related characteristics of pregnant women

The mean gestational age was 22.81 weeks (SD ± 8.3) and 134 (43.1%) of the pregnant women were in the second trimester and 117 (37.6%) were in their third trimester of pregnancy. Almost three forth 238(74.9%) of the pregnant women were multigravida while 78 (25.1%) were primigravida. Of the multigravida pregnant women, more than half of them 184 (59.2%) had an inter-pregnancy gap of more than two years while 49(15.8%) of them had an inter-pregnancy gap of less than or equal to two years (Table 2).

### Nutrition-related characteristics of pregnant women

The nutritional status of the participants measured using (MUAC) shows that nearly two-thirds, (64.3%) of the pregnant women had MUAC value within the normal limit (≥23 cm); conversely, 111(35.7%) of the participants had MUAC of <23 cm. Almost half, 155(49.8%), of pregnant

**Table 1. Socio-demographic and economic characteristics of pregnant women in southeastern zone, Tigray, Ethiopia, 2020 (n = 311).**

| Variable | Categories | Frequency (%) |
|---|---|---|
| Age (years) | 18–24 | 118 (37.9%) |
| | 25–31 | 123 (39.5%) |
| | 32–38 | 70 (22.5%) |
| Religion | Orthodox | 293 (94.2%) |
| | Muslim | 18 (5.8%) |
| Marital status | Married | 288 (92.6%) |
| | Single | 17 (5.5%) |
| | Divorced | 5 (1.6%) |
| | Widowed | 1 (0.3%) |
| Occupation | House wife | 135 (43.4%) |
| | Government employee | 19 (6.1%) |
| | Merchant | 34 (10.9%) |
| | Daily laborer | 5 (1.6%) |
| | Unemployed | 11 (3.5%) |
| | Farmer | 107 (34.4%0 |
| Family size | <5 | 185 (59.5%) |
| | ≥5 | 126 (40.5%) |
| Residence | Urban | 142 (45.7%) |
| | Rural | 169 (54.3%) |
| Monthly income | < 1000 | 73 (23.5%) |
| | 1000–2575 | 146 (46.9%) |
| | ≥ 2575 | 92(29.6%) |

**Table 2. Pregnancy-related characteristics of pregnant women attending ANC in southeastern zone, Tigray, Ethiopia, 2020 (n = 311).**

| Variables | Categories | Frequency (%) |
|---|---|---|
| Type of current pregnancy | Unplanned and Unwanted | 4 (1.3) |
| | Unplanned but wanted | 55 (17.7) |
| | Planned and Wanted | 252 (81) |
| Interpregnancy gap | Primigravida | 78 (25.1) |
| | $\leq$ 2 years | 49 (15.8) |
| | >2 years | 184 (59.2) |
| Gestational age | 1st trimester | 60(19.3) |
| | 2nd trimester | 134(43.1) |
| | 3rd trimester | 117(37.6) |
| Previous history of ANC follow up | Yes | 177 (56.9) |
| | No | 134 (43.1) |
| Family planning utilization before pregnancy | Yes | 181 (58.2) |
| | No | 130 (41.8) |
| Types of contraceptive use(n = 181) | Oral pills | 13 (7.2%) |
| | Injectable | 108 (59.7) |
| | Implanon | 54 (29.8) |
| | Jadelle | 4 (2.2) |
| | IUCD | 2 (1.1) |
| Had excessive menstrual bleeding | Yes | 7 (2.3) |
| | No | 304 (97.7) |
| Recent history of abortion | Yes | 58 (18.6) |
| | No | 253 (81.4) |
| Frequency of abortion | Once | 45 (77.6) |
| | $\geq$ 2times | 13 (922.4) |

women had taken iron-folic acid supplementation during their current pregnancy but more than one-third, 55(35.5), of them had low adherence to IFA supplementation (Table 3).

## Disease-related characteristics of pregnant women

A total of 285 (91.6%) pregnant women had no history of malaria attack in the past six months. Nearly all 307(98.7%) of the pregnant women reported they did not experience any acute illness. Besides only 6 (1.9%) of the pregnant women reported they are suffering from chronic disease (Table 4).

## Prevalence of anemia

Being adjusted for altitude the mean (±) SD of hemoglobin concentration was 11.9 g/dl (±1.35) with a range of 6.4g/dl to 16.8 g/dl. The overall prevalence of anemia among pregnant women who were attending their ANC follow-up in south Eastern zone of Tigray region was 55(17.7%) (95% CI; 13.2, 22.2). Regarding the severity of anemia, 34(61.8%) of the anemic cases showed a mild type of anemia while 18 (34.5%) and 3 (3.7%) of them had a moderate and severe type of anemia respectively (Fig 1).

## Contribution of iron deficiency anemia to total anemia

The median serum ferritin concentration among the anemic pregnant women was 15 (4.3–38.6). Of the 55 anemic pregnant women, 9(16.4%) of them had inflammation or infection

**Table 3. Nutrition-related characteristics of pregnant women in southeastern zone, Tigray, Ethiopia, 2020 (n = 311).**

| Variables | Categories | Frequency (%) |
|---|---|---|
| Meal frequency | ≤ 3 | 229 (73.6) |
| | >3 | 82 (26.4) |
| Coffee/tea intake immediately after food | Yes | 207 (66.6) |
| | No | 104 (33.4) |
| Fasting during pregnancy | Yes | 164 (52.7) |
| | No | 147 (47.3) |
| Nutrition counseling | Yes | 126 (40.5) |
| | No | 185 (59.5) |
| Knowledge about anemia | Good knowledge | 87 (28.0) |
| | Poor knowledge | 224 (72.0) |
| IFA supplementation | Yes | 155(49.8) |
| | No | 156(50.2) |
| Adherence to IFA supplementation(n = 155) | Low | 55 (35.5) |
| | Medium | 63 (40.6) |
| | High | 37 (23.9) |
| Nutritional status(MUAC) | < 23 | 111(35,7) |
| | ≥ 23 | 200(64.3) |
| Diet diversity score | Low | 233 (74.9) |
| | High | 78 (25.1) |
| Level of food insecurity | Food secured | 151 (48.6) |
| | Mildly food insecure | 127 (40.8) |
| | Moderately food insecure | 32 (10.3) |
| | Severely food insecure | 1 (3.0) |
| Foods/drinks prohibited by culture | Yes | 54 (17.4) |
| | No | 257 (82.6) |
| Food/drink items prohibited(n = 54) | Mustard | 36 (66.7) |
| | "Areki'(local alcoholic drink) | 9 (16.7) |
| | Roasted grain | 4 (7.4) |
| | Others(roasted grain, porridge) | 5 (.9.2) |
| Foods/drinks personally omitted | Yes | 23 (10.6) |
| | No | 278 (89.4) |

with high serum CRP levels (>5 mg/l). Out of 55(17.7%), anemic pregnant women 33(60%) of them had depleted iron stores (serum ferritin < 30 ng/ml) while 30(54.5%) of them had iron deficiency anemia (serum ferritin < 15 ng/ml) after adjusting the serum ferritin concentration for inflammation based on the WHO recommendation. These results indicate that more than half 54.5%, of anemia, observed among pregnant women was due to iron deficiency anemia, and the anemic pregnant women with low hemoglobin but normal ferritin values were considered to have anemia from other causes than iron deficiency.

## Factors associated with anemia

To determine possible associations of explanatory variables with anemia bivariate logistic regression analysis was applied. Hence, age of the pregnant women, residence, family income, family size, gestational age, inter-pregnancy interval, number of children, history of ANC follow-up, meal frequency, fasting during pregnancy, coffee/tea consumption immediately or within 1 hour after a meal, iron-folic acid supplementation, knowledge about anemia, DDS

**Table 4. Disease-related characteristics of pregnant women in southeastern zone, Tigray, Ethiopia, 2020 (n = 311).**

| Variables | Categories | Frequency (%) |
|---|---|---|
| History of malaria attack | Yes | 26 (8.4) |
| | No | 285 (91.6) |
| Current utilization of ITN | Yes | 182 (58.5) |
| | No | 129(41.5) |
| History of infection with an intestinal parasite | Yes | 17 (5.5) |
| | No | 294 (94.5) |
| Deworming (252) | Yes | 7 (2.8) |
| | No | 245 (97.2) |
| History of any acute illness | Yes | 4 (1.3) |
| | No | 307 (98.7) |
| Presence of chronic disease | Yes | 6 (1.9) |
| | No | 305 (98.1) |

and MUAC were associated with anemia at a p-value below 0.25. However, after adjusting for potential confounders the multivariable logistic regression analysis revealed that variables that were independent and significant predictors of anemia among the pregnant women were iron-folic acid supplementation, coffee/tea consumption immediately or within 1 hour after a meal, MUAC, gestational age, DDS, and interpregnancy interval.

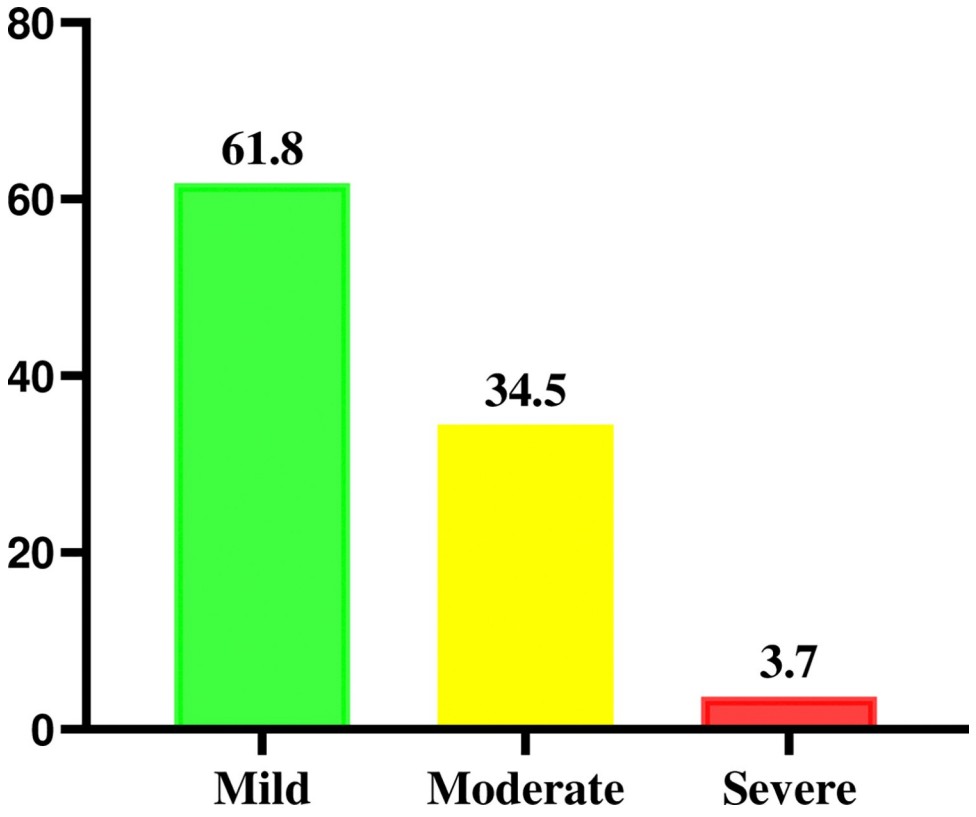

**Fig 1. Severity of anemia among pregnant women in south eastern Tigray, Ethiopia, 2020.**

The odds of having anemia among pregnant women who did not take iron-folic acid supplementation were 2.6 (AOR = 2.6, 95% CI: 1.045–6.70) times higher compared to those who took iron-folic acid supplementation during their pregnancy. Besides, the odds of anemia were 3(AOR = 3.0, 95% CI: 1.02–8.93) times higher in pregnant women who drink coffee/tea immediately or within 1 hour after a meal than their counterparts. The gestational age of the mothers was also significantly associated with anemia; increased odds of anemia were observed among pregnant women who were in their third trimester (AOR = 4.9, 95% CI: 1.49–15.98) than pregnant women who were in their first trimester. Besides, the odds of anemia were 2.7 (AOR = 2.7,95% CI: 1.16–6.48) higher in pregnant women with MUAC of less than 23 compared to pregnant women with MUAC of greater than or equal to 23. Diet diversity score was also found to be statistically significant with anemia among pregnant women; the odds of experiencing anemia among pregnant women with low DDS were 4.2 (AOR = 4.2, 95% CI: 1.018–17.58) times higher than the pregnant women with high DDS. Furthermore, the odds of anemia were 7.7(AOR = 7.7, 95% CI: 2.71–21.64) times more in pregnant women with less than or equal to two years interpregnancy gap than pregnant women who have delayed their birth interval for more than two years (Table 5).

## Discussion

In this study, the overall magnitude of anemia among pregnant women was found to be 17.7% (95% CI: 13.2–22.2). More than half of anemic cases were mild type (61.8%) followed by moderate (34.5%) and severe (3.6%) anemia respectively. The contribution of iron deficiency anemia to total anemia was 54.5%. Not taking IFA supplementation, low DDS, gestational age, MUAC of <23, pregnancy interval ≤ 2 years, and drinking coffee/tea after meals were significantly associated with anemia.

The magnitude of anemia in pregnant women in this study area is in line with studies done in Kefa zone19% [20], Cameron22% [21], Central zone of Tigray16.8% [22], and Tanzania18% [23]. The findings of this study were higher than studies done in northwest Ethiopia [14], Uganda [24], and Addis Ababa [12] where the prevalence of anemia was 10.6%, 7.4%, and 4.8% respectively. It's worth noting that the lower prevalence in northwest Ethiopia could be because only health institutions with the necessary materials to measure Hg were included in the study, whereas the lower prevalence in Addis Ababa may be attributed to the town's urban nature where mothers might have better nutrition awareness and access to nutritional diet. Moreover, the result was lower than a study done in Adama [25], Southern Ethiopia [26], Zambia [27], and Sudan [28], it was also lower when compared with the national prevalence of anemia reported in EDHS 2016 [9]. The possible reasons for the lower magnitude of anemia in the current study might be due to a significant proportion of participants 226 (72.7%) reported having received ANC follow-up, while almost half 155 (49.8%) reported taking IFA supplements. Additionally, the reduced prevalence of malaria in the study region, along with various socio-demographic and feeding behaviors of the participants, could also play a role in the observed differences in results.

According to this study, IDA was found to contribute to 54.5% (95% CI: 41.8–67.3) of total anemia in the study area, which aligns with the WHO's assumption about IDA's role in total anemia. However, the finding of this study was higher than a study done in rural Bangladesh [29], Singapore [30], and Uganda [31] in which the contribution of IDA to total anemia was 39%, 9.2%, and 23% respectively; but lower than a study done in Palestine in which the occurrence of all anemia was attributed to IDA [32]. These variations in results could be attributed to socioeconomic status, the use of IFA supplementation, the prevalence of other anemia-causing factors, and different cut-off points of serum ferritin to define anemia, along with the use of CRP to adjust the impact of ferritin limitation.

**Table 5. Bivariate and multivariable logistic regression analysis of factors associated with anemia among pregnant women in southeastern zone, Tigray, Ethiopia, 2020 (n = 311).**

| Variable | Category | Anemia status(n) | | COR(95% CI) | AOR(95% CI) |
|---|---|---|---|---|---|
| | | No | Yes | | |
| Age | 18–24 | 104 | 14 | 1 | |
| | 25–31 | 103 | 20 | 1.442 (0.691–3.009) | |
| | 32–38 | 49 | 21 | 3.184(1.494–6.785) * | |
| Residence | Urban | 126 | 16 | 1 | |
| | Rural | 130 | 39 | 2.362(1.257–4.442) * | |
| Monthly family income | <1000 | 51 | 22 | 3.176(1.421–7.098) * | |
| | 1000–2575 | 124 | 22 | 1.306(0.601–2.839) | |
| | >2575 | 81 | 11 | 1 | |
| Family size | <5 | 163 | 22 | 1 | |
| | ≥5 | 93 | 33 | 2.629(1.448–4.774) * | |
| Gestational age | 1st trimester | 54 | 6 | 1 | 1 |
| | 2nd trimester | 125 | 9 | 0.648(0.220–1.910) | 0.372(0.101–1.376) |
| | 3rd trimester | 77 | 40 | 4.675(1.852–11.80) * | 4.874(1.487–15.979) * |
| Pregnancy interval | Primigravida | 67 | 11 | 1.724(0.76–3.907) | 2.221(0.045–109.813) |
| | ≤ 2 | 22 | 28 | 13.2(6.224–28.355) ** | 7.663(2.713–21.644) ** |
| | > 2 | 167 | 16 | 1 | 1 |
| Number of children | Nulli para | 72 | 12 | 1 | |
| | Primi para | 57 | 7 | 0.737(0.272–1.993) | |
| | Multipara | 89 | 17 | 1.146(0.514–2.555) | |
| | Grand multipara | 38 | 19 | 3.000(1.318–6.830) * | |
| ANC | Yes | 198 | 28 | 1 | |
| | No | 58 | 27 | 3.298(1.799–6.024) ** | |
| Meal frequency | ≤3 | 182 | 47 | 2.389(1.077–5.299) * | |
| | >3 | 74 | 8 | 1 | |
| Fasting during pregnancy | Yes | 123 | 41 | 3.167(1.646–6.092) ** | |
| | No | 133 | 14 | 1 | |
| Coffee/ tea | Yes | 162 | 45 | 2.611(1.257–5.423) * | 3.022(1.023–8.927) * |
| | No | 94 | 10 | 1 | 1 |
| IFA supplementation | Yes | 136 | 19 | 1 | 1 |
| | No | 120 | 36 | 2.147(1.170–3.943) * | 2.646(1.045–6.703) * |
| Knowledge about anemia | Not knowledgeable | 179 | 45 | 1.936(0.928–4.039) * | |
| | Knowledgeable | 77 | 10 | 1 | |
| MUAC | <23 | 85 | 26 | 1.804(1.000–3.253) * | 2.749(1.166–6.477) * |
| | ≥23 | 171 | 29 | 1 | 1 |
| DDS score | Low | 181 | 52 | 7.182(2.175–23.715) * | 4.228(1.0177–17.577) * |
| | High | 75 | 3 | 1 | 1 |

COR = Crude Odds Ratio, AOR = Adjusted Odds Ratio, C.I = Confidence Interval; *(p<0.05); ** (p< = 0.001): reference group MUAC <23 = undernourished MUAC ≥ 23 = well-nourished DDS < 5 = poor, DDS≥ 5 = high adherence to IFA score of < 6 = low adherence, score of 6&7 = medium adherence and score of 8 = high adherence.

In this study, IFA supplementation was found to have a significant association with anemia in which pregnant women who did not take IFA supplementation were 2.6 times more likely to develop anemia as compared to the pregnant women who have had IFA supplementation. This might be due to the fact that, the demand for iron increases in pregnancy because of the need for the growing fetus, and placenta, and to sustain the associated biological changes;

hence, this increased demand is covered mostly by maternal iron stores which is difficult to cover by diet alone; for this reason, the pregnant women who have had IFA supplementation can increase their hemoglobin level and prevent anemia [4]. This result was supported by a study done in northwestern zone of Tigray in which the odds of anemia were 2.8 times higher among pregnant women who did not take iron supplementation during pregnancy [33]. Incongruent to this, a study done in southern Ethiopia also reveals that pregnant women who did not take IFA were 1.7 times more likely to be anemic [13]. Consistent evidence from a study done in Kenya also indicates that the odds of anemia were 2 times higher among mothers who did not take IFA supplementation [34].

This study also elucidates a significant and robust association between dietary diversity score (DDS) and the incidence of anemia among pregnant women. Specifically, mothers with low DDS exhibit a noteworthy 4.2-fold higher likelihood of developing anemia in comparison to their counterparts with high DDS. This finding was consistent with a study conducted in southern Ethiopia, where pregnant women with low DDS were found to be 3.18 times more at risk of developing anemia [13]. Another study conducted in Mekelle also supported this, revealing that pregnant women with low DDS were 13 times more likely to have anemia [35]. This might be due to the failure in meeting the increased demand for iron in pregnancy because of inadequate intake of iron-rich foods and most of the participants in this study were dependent on plant-based foods in which the iron absorption is poor. Furthermore, several substances in the diet can interfere with iron absorption since the staple food in the study area was cereals, the Phytates found in cereals can affect the bioavailability of iron [17].

This study also showed that mothers with inter inter-pregnancy interval of ≤ 2 years were 7.7 times more likely to develop anemia than mothers with an inter-pregnancy gap > 2 years. This finding was well-matched with a study done in Shire wherein mothers with an inter-pregnancy gap of less than two years displayed a 7.3-fold elevated risk of anemia relative to those with a gap of 2–3 years [36]. In support of this, evidence from a study done in Bangladesh revealed that the odds of having anemia were 2.7 times higher in pregnant women with birth spacing ≤2 years [37]. Moreover, a study done in Gamo Gofa also concludes that mothers with a pregnancy gap of ≤ 2 years were 3.1 times more likely to be anemic than those who had in pregnancy gap of > 2 years [38]. This can be related to the fact that pregnant women with frequent pregnancies might not restore the needed maternal nutrients that were depleted in their last pregnancy.

The level of trimester was also significantly associated with anemia. Compared to those who were in their first trimester, anemia was 4.9 times higher in pregnant women who were in their third trimester. This is congruent to a study done in Ghana where pregnant women in their third trimester displayed a fourfold increased risk of anemia in comparison to those in the first trimester [39]. In line with this, increased odds of anemia were observed in pregnant women who were in their third trimester according to a study done in Adigrat [40]. Furthermore, the findings from a study conducted in Bangladesh [37] provided additional support for the association between the third trimester of pregnancy and an augmented susceptibility to anemia. This might be due to the fact that physiologic expansion of maternal plasma volume, and the fetus and uterine content's requirement peak in the third trimester of gestation; late initiation of ANC might also contribute to the higher rate of anemia in the third trimester.

This study has also found a significant correlation between anemia and the MUAC of pregnant women. In particular, those with a MUAC value of less than 23 cm were 2.8 times more likely to be anemic than those with a MUAC of 23 cm or greater. This finding aligns with prior investigations conducted in Dera, wherein mothers with MUAC values below 23 cm were found to be 4.9 times more prone to anemia [41]. This was also supported by a study done in JigJiga in which pregnant women with MUAC measurements of 23 cm or greater exhibited a

57% reduced likelihood of anemia compared to those with MUAC below 23 cm [42]. Furthermore, a concordant study conducted in Kenya [34] corroborated these findings. This might be due to the fact that pregnant women with low nutritional status have a high probability of being micronutrient deficient hence, they can develop anemia as a result of that.

This study also revealed that the highest likelihood of anemia was observed in pregnant women who drink coffee/tea immediately after a meal or within a 1-hour gap; they were 3 times more likely to be anemic as compared to their counterparts. In agreement with this finding, a study done in Kefa zone found that pregnant women who drank coffee or tea after a meal were twice as likely to experience anemia as those who did not consume these beverages [20]. Concordant with this finding, a study conducted in Adama found that pregnant women who drank coffee were 5.5 times more likely to develop anemia [25]. This may be due to the caffeine in coffee and the tannins in tea, which can impede the absorption of non-heme iron [43].

The fact that the other factors in this study such as:—maternal age, parity, educational status, malaria, HIV, and heavy menstrual bleeding failed to be associated with anemia might be explained by the high percentage of literacy in the current study, low incidence of HIV in the study area, different methods used to assess menstrual bleeding and no endemic of malaria in the current study.

As strength of the study primary data was collected with a high response rate. The hemoglobin level of each pregnant woman was determined in the field. Serum ferritin which is the gold standard biochemical marker with high specificity was used to assess the iron status of anemic pregnant women; C reactive protein was also measured to control the effect of inflammation on ferritin determination.

As a limitation of this study cross-sectional study was utilized in this study which cannot measure the cause-and-effect relationship between the factors and outcome variable. The other potential limitations of this study are recall bias, and social desirability bias while the pregnant women were requested to give dietary information they consumed on the preceding day (24-hour recall method) and monthly income of their family.

## Conclusion

In conclusion, the study revealed that 17.7% of pregnant women in the southeastern zone of the Tigray region experience anemia, which is considered a minor public health issue. The majority of anemia cases were classified as mild, with 54.5% attributable to IDA. Factors found to be significantly and independently associated with anemia during pregnancy include a gap of ≤ 2 years between pregnancies, third trimester gestational age, consumption of coffee/tea immediately or within an hour after a meal, lack of IFA supplementation, low dietary diversity score, and a MUAC value of < 23. To reduce anemia, pregnant women should be counseled on consuming iron-rich foods, diversifying their diet, increasing meal frequency, and delaying coffee/tea consumption that inhibits iron absorption. Healthcare providers should also counsel on the importance of taking IFA supplementation, early ANC follow-up, and, importance of using family planning to reduce the burden of anemia caused by frequent pregnancies. Furthermore regional and federal ministries of health should work on iron fortification to address anemia at large.

## Supporting information

**S1 Data.**
(SAV)

## Acknowledgments

Sincere gratitude be to Mekelle University for the invaluable support they provided throughout the study. We would also like to extend our heartfelt appreciation to the participants of the study, as well as the data collectors, and supervisors, for their cooperation.

## Author Contributions

**Conceptualization:** Tedros Bereket, Freweini Gebrearegay Tela, Gebretsadkan Gebremedhin Gebretsadik, Selemawit Asfaw Beyene.

**Data curation:** Tedros Bereket, Freweini Gebrearegay Tela, Gebretsadkan Gebremedhin Gebretsadik, Selemawit Asfaw Beyene.

**Formal analysis:** Tedros Bereket.

**Funding acquisition:** Tedros Bereket.

**Investigation:** Tedros Bereket, Freweini Gebrearegay Tela, Selemawit Asfaw Beyene.

**Methodology:** Tedros Bereket.

**Project administration:** Tedros Bereket.

**Resources:** Tedros Bereket.

**Software:** Tedros Bereket.

**Supervision:** Tedros Bereket, Freweini Gebrearegay Tela, Selemawit Asfaw Beyene.

**Validation:** Tedros Bereket, Gebretsadkan Gebremedhin Gebretsadik.

**Visualization:** Tedros Bereket, Freweini Gebrearegay Tela, Gebretsadkan Gebremedhin Gebretsadik, Selemawit Asfaw Beyene.

**Writing – original draft:** Tedros Bereket, Freweini Gebrearegay Tela, Selemawit Asfaw Beyene.

**Writing – review & editing:** Tedros Bereket, Freweini Gebrearegay Tela, Gebretsadkan Gebremedhin Gebretsadik, Selemawit Asfaw Beyene.

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
