## [Decision Letter · Decision Letter 0]

8 Aug 2024

PONE-D-23-33484Contribution of iron deficiency to anemia and Factors associated with anemia during pregnancy in Southeastern Tigray, Ethiopia, 2020PLOS ONE

Dear Dr. weldejewergs,

Thank you for submitting your manuscript to PLOS ONE. After careful consideration, we feel that it has merit but does not fully meet PLOS ONE’s publication criteria as it currently stands. Therefore, we invite you to submit a revised version of the manuscript that addresses the points raised during the review process.

We look forward to receiving your revised manuscript.

Kind regards,

Tebelay Dilnessa, MSc

Academic Editor

PLOS ONE

4. We notice that your supplementary figures are uploaded with the file type 'Figure'. Please amend the file type to 'Supporting Information'. Please ensure that each Supporting Information file has a legend listed in the manuscript after the references list.

Additional Editor Comments:

Line 25: Therefore, this study was aimed at assessing the contribution……..’ It is better written as, ‘Therefore, this study aimed to assess the contribution………’Line 28-30: ‘…….was conducted on 311 pregnant women who were attending four health facilities in the southeastern zone of Tigray region between January 2020 and June 2020.’ It is better written as, ‘………was conducted among 311 pregnant women who attended four health facilities in the southeastern zone of Tigray region from January 2020 to June 2020.’What about ‘Bivariate logistic regression’ and ‘Multivariate logistic regression’ in the abstract?In the abstract, the p-value should be included.
In the abstract and result, the absolute number (numerator and denominator) is needed together with the percentage. For example, A/B (17.7%).
Line 30: ……..to collect data.’ It is better written as, ‘………to collect the data’
The introduction requires revision. The rationale is not explained well. Why you did this research? What has been done before, what is known before and which you intend to fill?
Line 100: Rewrite as, ‘Materials and Methods’Line 102: The study was conducted among 311 pregnant women from January 2020 to June 2020….Line 112: Rewrite as, ‘Study design’, because the ‘setting’ is already described in the study area, period and population.Line 140: Rewrite as, Data collectionLine 158: Better be, ‘Laboratory Methods’The ‘data quality assurance’ portion should be rewritten. Select the most important and present.Revision was need in the’ data analysis’ section. Binary? Do you want to say ‘Bivariate logistic regression’ and ‘Multivariate logistic regression?’ Additionally, the statement ‘First, descriptive statistics of the study variables were done and data were presented using tables, and figures’ should be removed.A separate heading ‘Conclusion’ is required. If there is any recommendation you can add here.Follow the guideline for manuscript writing protocol for PLoS One.

Reviewers' comments:

Reviewer's Responses to Questions

**Comments to the Author**

1. Is the manuscript technically sound, and do the data support the conclusions?

Reviewer #1: Yes

Reviewer #2: Yes

2. Has the statistical analysis been performed appropriately and rigorously? 

Reviewer #1: No

Reviewer #2: Yes

3. Have the authors made all data underlying the findings in their manuscript fully available?

Reviewer #1: Yes

Reviewer #2: Yes

4. Is the manuscript presented in an intelligible fashion and written in standard English?

Reviewer #1: No

Reviewer #2: No

5. Review Comments to the Author

Reviewer #1: I read the paper with interest and it was really good, however there are area need to improve and required further explanation.

Introduction part

Some of your data in the introduction part was old enough kindly let you replace it with the new information. The claim that contribution of anemia is unknown looks like exaggerated. we know many study conducted in Ethiopia and in Tigray region too.

Methods.

The study setting for this particular study was not clear? Your study design was facility based cross sectional study however in which particular setting that the study conducted was not noted

Your study population need to be well specific. It stated “pregnant women “ …..it was better if you clearly specified those pregnant women and from where you recruited them? Do you mean pregnant women come for ANC? Or pregnant women come hospital for other purpose? It recommended if you well specified your study population separately.

Like that of study population, your study failed to specified the exact inclusion criteria.

Sample size determination you used was from the EDHS 2016 which was population based survey, why you didn’t used the recent study with similar study population

The calculated sample size was 317. and why not considered the non-response rate. it is common to use include about 10 %or 5% the sample size as non-response rate.

The sampling procedure was not clear. Need to specified each. Put number for each steps.

Your study variable is contribution of anemia or anemia???

The tools for data collection should be cited eg DDS

How you check the validity of Hem cue measurement. what was its principle. please specify amount of blood you collected by cuvette \\

It stated “Finally the punctured part was wiped for any remaining blood from the third drop and covered with an alcohol swab.” Do you think that was simple acceptable you covered that punctured site with alcohol swap??

How you measure adherence to IFA and the knowledge was not clear

What you did for those anemic pregnant women

Result

“Number of ANC Visit” the word visit should replaced by contact

In table to of the contraceptive it says “Jagule” I never encounter such named contraceptive kindly specify it

It recommended that that px women should take meal about four time in day however in your case about three-fourth of participants get food less than three time. Could you please justify it?

Only about 28% of your participant had good knowledge of anemia? while many of them were previously attended ANC and 40% had nutritional counseling. Please, how you justified this finding?

It was not clear the way you measure food security, again you didn’t mention it in the variables

How you compare food diversity and food security result in your study? Please check it again??

Most of your study participants take meal less than three time per day. again 151 participants were food secured. how you justify it??

About 17 women infected with I parasite but only 7 were deworming??

Please specified those chronic diseases and the FP utilization time

It said “The odds of having anemia among pregnant women who did not take iron-folic acid supplementation were 2.6time.” but no figure show number of px women who did not supplemented IFA?

DDS and inter pregnancy interval show big odds ratio with wide CI. do you think it was recommended to have such result?

In table 2 the Previous history of ANC follow up was reported by 177 but in table 5 about 226 women had ANC contact indicating that the result was manipulated. similarly check for pregnancy interval. look also it say in the written part 155 women supplemented IFA but no figures a specified in table. At the same time MUAC was not reported in the table of the descriptive. there are many variables not reported in the disruptive tables but show in the table 5

The manuscript have no recommendation at all

Reviewer #2: Comments to the author

Although the paper has some interesting sections, there are shortcomings that need to be fixed.

General: The manuscript has some grammatical errors that need to be fixed. For example, lines 70 and 499. It also needs formal writing, which must be different from a thesis. 

Introduction:

Some polishing is needed on the grammar.

Integrating lines 59–62 with other sections would be preferable.

Methods:

The study design and setting should be combined with the study area section.

Words like "first cuvette"- is there a second cuvette for one subject? "third drop," etc. are inappropriate in this context, so lines 159–166 must be rewritten.

A brief, succinct, and formal writing style is required in the data quality assurance section.

Discussion:

There are a few grammar mistakes that should be fixed. Consider line 499.

6. PLOS authors have the option to publish the peer review history of their article (what does this mean?). If published, this will include your full peer review and any attached files.

Reviewer #1: **Yes: **Abdurahman Kedir Roble

Reviewer #2: **Yes: **Ayenew Assefa

---

## [Author Response · Author response to Decision Letter 0]

24 Sep 2024

Point by point response to reviewers’ comments

Manuscript number: PONE-D-23-33484 

Dear PLOSE ONE Editor, 

We are grateful for the opportunity to submit a revised draft of our manuscript titled “Contribution of iron deficiency to anemia and Factors associated with anemia during pregnancy in Southeastern Tigray, Ethiopia, 2020: A cross-sectional study” to PLOSE ONE. We would like to express our gratitude for the time and effort you and the reviewers have dedicated to providing valuable feedback on our manuscript. All the insightful comments and feedbacks from the editor has been corrected accordingly. The insightful comments from the reviewers have also been greatly appreciated, and we have endeavored to incorporate all of the suggestions provided. All changes made to the manuscript have been carefully tracked. 

Below, you will find a detailed response to the reviewers’ comments. 

Reviewer comment Response 

 Reviewer 1

1. Some of your data in the introduction part was old enough kindly let you replace it with the new information. The claim that contribution of anemia is unknown looks like exaggerated. we know many study conducted in Ethiopia and in Tigray region too. 

 Dear reviewer, as per your suggestion, we have replaced the outdated information in the introduction with the updated one. Regarding the comment that mentioned the claim about the contribution of anemia being unknown seemed exaggerated, our rationale for that statement was the lack of research in the country assessing both iron deficiency and anemia. However, we have now revised it to "limited studies" instead of "unknown." (Line 90)

methods

A. The study setting for this particular study was not clear? Your study design was facility based cross sectional study however in which particular setting that the study conducted was not noted 

 Dear reviewer, the study setting for this study was south eastern zone, and four health facilities that are found in south eastern zone were randomly selected to conduct the study among pregnant women who came for ANC follow-up. We have made adjustments to enhance the readability of the text. (Line 96)

B. Your study population need to be well specific. It stated “pregnant women “ …..it was better if you clearly specified those pregnant women and from where you recruited them? Do you mean pregnant women come for ANC? Or pregnant women come hospital for other purpose? It recommended if you well specified your study population separately.

 Dear reviewer, we have modified that accordingly. 

C. Like that of study population, your study failed to specified the exact inclusion criteria. 

 We have made the necessary amendment accordingly.(Line 108-112) 

D. Sample size determination you used was from the EDHS 2016 which was population based survey, why you didn’t used the recent study with similar study population 

 Dear reviewer, we used this large survey because there were limited studies that assessed both iron deficiency and anemia among pregnant women. Moreover, we wanted to get a larger sample size powerful enough to answer our research questions and detect any potential significant associations.

E. The calculated sample size was 317. and why not considered the non-response rate. it is common to use include about 10 %or 5% the sample size as non-response rate

 Dear reviewer, thank you for your valuable insight. We realized that we failed to explain how we arrived at the final sample size. Here's the explanation: Initially, the calculated sample size was 318. However, due to the lower number of pregnant women with ANC follow-up in the southeastern zone (according to the 2018/2019 report), we applied a correction formula, which resulted in a final sample size of 288. Finally by considering 10% non-response rate the final sample size was 317.(Line 114-120)

F. The sampling procedure was not clear. Need to specified each. Put number for each steps

 Dear reviewer, we have modified and made it specific. We also have put number for each selected health facility accordingly.

G. Your study variable is contribution of anemia or anemia??? Our dependent study variable is anemia.

H. The tools for data collection should be cited eg DDS

 Dear reviewer, we have cited the references accordingly. (Line 150) 

I. How you check the validity of Hem cue measurement. what was its principle. please specify amount of blood you collected by cuvette

 Dear reviewer, we verified the validity of the Hem cue measurement by using the precalibrated hem cue measurement and following Standard Operating Procedures (SOPs) and the manufacturer’s instructions for laboratory activities. We always ensure that the analyzer is in the loading position and check for the three flashing dashes, which indicate that the analyzer is ready for use. Additionally, we clean the cuvette holder each day with 70% alcohol and ensure that it is completely dry before re-inserting it.

 A drop of capillary blood (10µl) Capillary blood was drawn, through micro cuvettes and inserted into the Hemo Cue Hb analyzer and the result were recorded. (Line 169)

J. It stated “Finally the punctured part was wiped for any remaining blood from the third drop and covered with an alcohol swab.” Do you think that was simple acceptable you covered that punctured site with alcohol swap??

 Dear reviewer, we were not suggesting that the procedure is just acceptable. Our intention was to demonstrate the blood collection procedures as outlined in the standard operating procedures (SOP) manual. This involves cleaning the fingertip (puncture site) with alcohol and providing a gauze pad to the patient to apply pressure on the puncture site for a minute in order to stop the bleeding.

K. How you measure adherence to IFA and the knowledge was not clear

 Dear reviewer, The Morisky Medication Adherence Score was used to assess the adherence of pregnant women to IFA supplementation. (Line 232-233)

To assess the knowledge of pregnant women about anemia, all the participants were asked ten questions related to anemia, accordingly, the pregnant women were grouped as having good knowledge if they scored greater than or equal to the mean value of correct responses and poor knowledge if they scored less than the mean value of correct responses. (Line 159-162)

L. What you did for those anemic pregnant women 

 Dear reviewer, When we encounter pregnant women with anemia, the first step we did was informing them about their anemia status and provide nutritional counseling. We also explain the potential implications of anemia for their health and their baby’s well-being. Following this, we ensure they are linked back to the ANC clinic to receive appropriate healthcare services, including iron folic acid supplementation and further evaluation and management.

Results 

1. “Number of ANC Visit” the word visit should replaced by contact 

 We have modified the word accordingly. 

2. In table to of the contraceptive it says “Jagule” I never encounter such named contraceptive kindly specify it 

 Dear reviewer, It was a typing error and we have modified it as Jadelle. 

3. It recommended that that px women should take meal about four time in day however in your case about three-fourth of participants get food less than three time. Could you please justify it?

 Dear reviewer, about three-fourth of participants get food less than or equal to three time, though it is recommended that pregnant women are advised to take a meal about four time a day they typically do not adhere to this recommendation due to the cultural practice of the Tigrian community, as they eat three meals a day (breakfast, lunch, and dinner).

4. Only about 28% of your participant had good knowledge of anemia? while many of them were previously attended ANC and 40% had nutritional counseling. Please, how you justified this finding?

 Dear reviewer, this could be explained due to the fact that nutritional counseling is often general and may not be comprehensive. It mainly focuses on dietary advice, it might not have specifically addressed anemia, plus, the mother’s level of understanding and ability to retain information may also have an impact. Moreover, the ANC care provider may not have enough time to address all nutrition-related issues.

5. It was not clear the way you measure food security, again you didn’t mention it in the variables

 Dear reviewer, as you suggested we have added it in data collection method how we measure food security. (Line 150-158)

Dear reviewer, we have mentioned food security in the independent variables list.(Line 134)

6. How you compare food diversity and food security result in your study? Please check it again??

 Dear reviewer, we used a 24-hour recall individual dietary diversity questionnaire for the diet diversity data and, the household food security score was utilized for the food security data. We have rechecked the result, and is correct.

7. Most of your study participants take meal less than three time per day. again 151 participants were food secured. how you justify it??

 Dear reviewer, this might be explained by the fact that the difference in the time frame for asking the two questions could potentially impact the responses given. Moreover, participants may have enough food but choose to eat fewer meals in order to manage their budget effectively

8. About 17 women infected with I parasite but only 7 were deworming?? 

 Dear reviewer, the difference in numbers between parasite infection and deworming may be due to some healthcare provider’s not offering deworming treatment if they do not see the infection as severe. Additionally they might delay deworming until later in pregnancy, following WHO recommendations, due to concerns about the safety of medications for the developing fetus.

9. Please specified those chronic diseases and the FP utilization time

 Dear reviewer, we have specified the time accordingly. The time for FP is before getting pregnant and the time for chronic illness is inclusive, asking for both before and during the current pregnancy.

10. It said “The odds of having anemia among pregnant women who did not take iron-folic acid supplementation were 2.6time.” but no figure show number of px women who did not supplemented IFA? 

 Dear reviewer, Thank you for your valuable insight, we realized that we neglected to include the variable for IFA intake during the current pregnancy, which was originally responded to as either 'Yes' or 'No'. We have since rectified this and included the variable in the results. (Table 3 row 6)

11. DDS and inter pregnancy interval show big odds ratio with wide CI. do you think it was recommended to have such result?

 Dear reviewer, we have rerun the analysis to ensure the findings and The results for both DDS and interpregnancy gap remain consistent .The odds ratio was also supported by other studies conducted in the country.

12. In table 2 the Previous history of ANC follow up was reported by 177 but in table 5 about 226 women had ANC contact indicating that the result was manipulated. similarly check for pregnancy interval. look also it say in the written part 155 women supplemented IFA but no figures a specified in table. At the same time MUAC was not reported in the table of the descriptive. there are many variables not reported in the disruptive tables but show in the table 5 

 Dear reviewer, thank you for identifying these discrepancies in the tables and pointing out the inconsistencies between the reported numbers. We have addressed these discrepancies by carefully reanalyzing the data. Upon review, we realized that we had omitted some important variables from the result tables. Specifically, we failed to include IFA supplementation, gestation age, and nutritional status (MUAC) in the initial results table. These variables have now been added to the results table. Additionally, we acknowledge the discrepancy in the number of ANC. After reanalyzing the data, we confirmed that a typing error had occurred during the insertion of the number in the results section. This error has been rectified accordingly. (Table 2 row 3, Table 3 row 6 and 8) 

13. The manuscript have no recommendation at all 

 Dear reviewer, Thank you for your valuable insights. The recommendation part was unintentionally omitted but we have rectified this by incorporating the recommendation along with the conclusion. (Line 457-463)

 Reviewer 2

1. General: The manuscript has some grammatical errors that need to be fixed. For example, lines 70 and 499. It also needs formal writing, which must be different from a thesis. Dear reviewer, Thank you for pointing out the grammatical errors in the manuscript. Following a comprehensive review, we have diligently rectified the inconsistencies and refined the language to ensure its utmost clarity. Moreover, we have refined the manuscript to a more formal writing style to align with the requisite standards of the journal.

2. Some polishing is needed on the grammar.

Integrating lines 59–62 with other sections would be preferable Dear reviewer, we have thoroughly addressed the grammatical errors and made an effort to integrate lines 59–62 with other sections, in accordance with your recommendations.

3. The study design and setting should be combined with the study area section.

Words like "first cuvette"- is there a second cuvette for one subject? "third drop," etc. are inappropriate in this context, so lines 159–166 must be rewritten. Dear reviewer, we have made the necessary modifications and rewritten the paragraph accordingly.

4. A brief, succinct, and formal writing style is required in the data quality assurance section Dear reviewer, we have made the necessary amendments for the data quality assurance section. (Line 192-205)

5. There are a few grammar mistakes that should be fixed. Consider line 499. Dear reviewer, thank you for your insight and we have made the necessary corrections for grammatical and punctuation errors accordingly.

We think we have addressed the points you have raised and we look forward to hearing from you in due time regarding our revised manuscript and to respond to any further questions and comments you may have.

Sincerely, 

Tedros Bereket (Corresponding author)

Email: Tedros.berekt99@gmail.com

---

## [Decision Letter · Decision Letter 1]

6 Nov 2024

PONE-D-23-33484R1The role of iron deficiency and Factors associated with anemia during pregnancy in Southeastern Tigray, Ethiopia, 2020PLOS ONE

Dear Dr. Bereket,

Thank you for submitting your manuscript to PLOS ONE. After careful consideration, we feel that it has merit but does not fully meet PLOS ONE’s publication criteria as it currently stands. Therefore, we invite you to submit a revised version of the manuscript that addresses the points raised during the review process.

We look forward to receiving your revised manuscript.

Kind regards,

Tebelay Dilnessa, MSc

Academic Editor

PLOS ONE

Journal Requirements:

Additional Editor Comments:The paper was significantly improved. But since there was no a point-by-point response to editor’s concern, I invite you again to see the previous commentsStudy variables should be written as ‘Dependent variable’ and ‘Independent variables’ clearly and separately. No description will be needed here, for example, 0 and 1.Previous comment: In the abstract and result, the absolute number (numerator and denominator) is needed together with the percentage. For example, A/B (17.7%). Make a correction or clarify it in the ‘Response to reviewers comment’.Line 252: Socio-demographic characteristics of study participantsWhy you leave P-value in the ‘associated factors’ assessment ‘table’?In the discussion part, the last three paragraphs were not supported by evidence/citation and they require revision.Write the ‘operational definitions’ to the standardPut the description of supplementary file (s) below the reference listsKeeping appropriate statements selected in the submission system, you can remove the following from the main manuscript.

**Consent for publication **

Not applicable

**Availability of data and materials **

All relevant data are found within the manuscript and raw data supporting the findings of this paper are available upon reasonable request from the corresponding author.

**Competing interest **

The authors declare that they have no competing interests.
The references were not written properly. All references should be written correctly according to Vancouver style. For example, reference number 17 can be written correctly as, Kenea A, Negash E, Bacha L, Wakgari N. Magnitude of Anemia and Associated Factors among Pregnant Women Attending Antenatal Care in Public Hospitals of Ilu Abba Bora Zone, South West Ethiopia: A Cross-Sectional Study. *Anemia.* 2018, 2018: 1-7.  Here the name of the journal was not *Hindawi*, rather it is *Anemia.*Use the alignment ‘Justify’ for the text throughout the manuscriptFollow the standard binomial nomenclature, italize journal name and the word ‘et al’

Reviewers' comments:

Reviewer's Responses to Questions

**Comments to the Author**

1. If the authors have adequately addressed your comments raised in a previous round of review and you feel that this manuscript is now acceptable for publication, you may indicate that here to bypass the “Comments to the Author” section, enter your conflict of interest statement in the “Confidential to Editor” section, and submit your "Accept" recommendation.

Reviewer #2: All comments have been addressed

2. Is the manuscript technically sound, and do the data support the conclusions?

Reviewer #2: Yes

3. Has the statistical analysis been performed appropriately and rigorously? 

Reviewer #2: Yes

4. Have the authors made all data underlying the findings in their manuscript fully available?

Reviewer #2: Yes

5. Is the manuscript presented in an intelligible fashion and written in standard English?

Reviewer #2: Yes

6. Review Comments to the Author

Reviewer #2: The authors make an effort to respond to the criticisms. As a last note, it would be preferable if the writers allotted sufficient time for editing and eliminating superfluous and pointless words from the document, if they exist.

7. PLOS authors have the option to publish the peer review history of their article (what does this mean?). If published, this will include your full peer review and any attached files.

Reviewer #2: **Yes: **Ayenrw Assefa

---

## [Author Response · Author response to Decision Letter 1]

12 Jan 2025

Dear reviewers, we have addressed all the comments that were given to us and explained point by point in the response to the reviewers and editors letter.

Thank you.

---

## [Editor Report · Decision Letter 2]

14 Jan 2025

The role of iron deficiency and Factors associated with anemia during pregnancy in Southeastern Tigray, Ethiopia, 2020

PONE-D-23-33484R2

Dear Dr. Bereket,

We’re pleased to inform you that your manuscript has been judged scientifically suitable for publication and will be formally accepted for publication once it meets all outstanding technical requirements.

Kind regards,

Tebelay Dilnessa, MSc

Academic Editor

PLOS ONE

Additional Editor Comments (optional):

The tile for 'Figure 1: Severity of anemia among pregnant women in southeastern zone, Tigray, Ethiopia, 2020' should be inserted below line 297 and the figure 1 should be prepared without title in tif version.
---

## [Editor Report · Acceptance letter]

19 Jan 2025

PONE-D-23-33484R2 

PLOS ONE

Dear Dr. Bereket, 

I'm pleased to inform you that your manuscript has been deemed suitable for publication in PLOS ONE. Congratulations! Your manuscript is now being handed over to our production team.

Kind regards, 

on behalf of

Dr. Tebelay Dilnessa 

Academic Editor

PLOS ONE